# Interface Interactions in Conjugated Polymer Composite with Metal Oxide Nanoparticles

**DOI:** 10.3390/nano9111534

**Published:** 2019-10-29

**Authors:** Tsegaye Belege Atisme, Chin-Yang Yu, Eric Nestor Tseng, Yi-Che Chen, Pei-Kai Hsu, Shih-Yun Chen

**Affiliations:** Department of Materials Science and Engineering, National Taiwan University of Science and Technology, 43, Section 4, Keelung Road, Taipei 10607, Taiwan; tsfre99@gmail.com (T.B.A.); cyyu@mail.ntust.edu.tw (C.-Y.Y.); ernestor88@gmail.com (E.N.T.); d10704001@mail.ntust.edu.tw (Y.-C.C.); D10804002@mail.ntust.edu.tw (P.-K.H.)

**Keywords:** dispersion, charge-transfer, fluorescence quenching, red-shifting, defects

## Abstract

This study presents the preparation, characterization, and properties of a new composite containing cerium oxide nanoparticles and a conjugated polymer. CeO_2_ nanoparticles prepared using the co-precipitation method were dispersed into the conjugated polymer, prepared using the palladium-catalyzed Suzuki–Miyaura cross-coupling reaction. The interface interactions between the two components and the resultant optoelectronic properties of the composite are demonstrated. According to transmission electron microscopy and X-ray absorption spectroscopy, the dispersion of CeO_2_ nanoparticles in the polymer matrix strongly depends on the CeO_2_ nanoparticle concentration and results in different degrees of charge transfer. The photo-induced charge transfer and recombination processes were studied using steady-state optical spectroscopy, which shows a significant fluorescence quenching and red shifting in the composite. The higher photo-activity of the composite as compared to the single components was observed and explained. Unexpected room temperature ferromagnetism was observed in both components and all composites, of which the origin was attributed to the topology and defects.

## 1. Introduction

Composites containing nanoparticles have received much attention due to their synergistic and hybrid properties derived from their corresponding components. The composite properties depend on the properties of their individual components and also on their morphologies and interfacial characteristics. For instance, Bellucci et al. demonstrated that the optical properties changes were correlated to nanoparticle-driven interface modulations [1]. The potential uses of composites include battery cathodes and solid-state ionics [2,3], supercapacitor and dielectrics [4,5], catalysts [6,7], corrosion protection [8,9], sensors [10], optical and electrical materials [11,12], ultraviolet (UV) blocking [13,14,15] and many others. In these recent years, composites with the use of organic–inorganic hybrids were found to possess improved and new properties. The main challenge in polymer-inorganic nanoparticles (NP) is cluster or agglomerate formation due mainly to the presence of strong Van Der Waals and electrostatic forces. In general, there are two preparative methods for these composites such as the in situ and the ex situ methods [16]. The in situ method prepares the nanoparticles in the polymer solution or monomer polymerization containing the nanoparticles. This method gives better nanoparticle dispersion but the problem is nanoparticle contamination with solids that should be removed. The ex situ method involves nanoparticle and polymer preparation separately before mixing them in a solvent using mechanical forces. This method is relatively simple and commonly used for large-scale production even if the dispersion is less efficient compared to the in situ method and difficult to maintain stable dispersion.

Among the rare earth metal oxides that have been used for composite fabrication, cerium (IV) oxide (ceria, CeO_2_) is one of the most used metal oxides because cerium is the most abundant element among the rare earths and is environmentally friendly [17]. An interesting property of CeO_2_ is that it can have a stable structure far from the stoichiometric proportions of oxygen. The relaxation or reconstruction of materials with defects at high concentrations is unusual. Both in theory and experimentally, it has been shown that the defects tend to concentrate at the surface, which leads to unique catalytic behavior as well as magnetism. T. Masui et al. worked the relaxatived auto-dispersion (RAD) process to prepare CeO_2_ nanoparticles dispersed in Nylon 11 film matrix [18]. They prepared CeO_2_ nanoparticles with an average size of 5 nm. The uniform dispersion of CeO_2_ nanoparticles in the polymer matrix was observed by the preparation of CeO_2_ from ultrafine cerium metal particles deposited upon nylon film in vacuum using oxidation in air. W. Wang et al. used a direct facile synthetic strategy which is the in situ type to incorporate CeO_2_ in regenerated cellulose films [19]. In their work, CeO_2_ was fairly uniformly dispersed in the polymer matrix with an average size of about 24 nm. When the CeO_2_ precursor concentration increased, the CeO_2_ nanoparticle agglomeration in the polymer also became higher. Their work showed that the prepared nanocomposites have moderate thermal stability, a certain degree of hydrophobicity and desired UV-shielding property. However, this method may not be suitable to control the nanoparticle shape and size.

Recently, F. C. da Silva et al. further demonstrated the charge transfer process between CeO_2_ and 1,4,5,8-naphthalenediimide [20]. In their work, they briefly explained that the photochemical property of the diimide molecules in the nanocomposites can be tuned at different pH values due to the sorption of molecules on the CeO_2_ nanoparticle surface. The diimide molecules adsorb onto the surface and are sensitive to the CeO_2_ surface charge. In more acidic conditions, the fluorescence channel of the naphthalenediimide is triggered, while at basic condition, the creation of charge transfer complexes is induced. Thus, finding a suitable technique for synthesizing a composite in which CeO_2_ nanoparticles are dispersed homogeneously in a polymer matrix is a fascinating task. Tuning the properties of the composites is necessary to investigate the interactions between polymers and oxides in more detail.

In this paper, a conjugated polymer containing 2,7-linked spirobiflourene and naphthalene bisimide was utilized to form a composite with CeO_2_ nanoparticles. An ex situ method was used to prepare the polymer composite with CeO_2_ nanoparticles. The polymer used in this study can accept electron from electrophilic species. On the other hand, CeO_2_ is a superior catalyst, which easily adjusts the surface condition. Nanoparticle dispersion, interactions between the components, structure and the physical properties of the composites are investigated and compared with that of the individual components.

## 2. Experimental Procedure

### 2.1. Synthesis of Polymer

The conjugated polymer was synthesized by the modification procedures reported previously [21]. All chemicals and reagents were purchased from commercial sources (TCI, Alfa Aesar (Haverhill, MA, United States) or Sigma Aldrich (St. Louis, MO, United States)) and used without further purification unless otherwise noted. The monomers are 2,7-bis(4,4,5,5-tetramethyl-1,3,2-dioxaborolan-2-yl)-9,9’-spirobifluorene, 4,7-dibromobenzo[*c*] thiadiazole and *N,N’*-bis(2-ethylhexyl)-2,6-dibromonaphthalene-1,4,5,8-tetracarboxylic acid diimide. Other chemicals used for polymer synthesis are aqueous potassium carbonate (K_2_CO_3_), tetrakis (triphenylphosphine) palladium and 1,4-dioxane. This alternating copolymer was obtained via Suzuki–Miyaura cross coupling reaction of one equivalent of 4,7-dibromobenzo[*c*] thiadiazole, two equivalents of 2,7-bis(4,4,5,5-tetramethyl-1,3,2-dioxaborolan-2-yl)-9,9’-spirobifluorene and one equivalent of *N,N’*-bis(2-ethylhexyl)-2,6-dibromonaphtalene-1,4,5,8-tetracarboxylic acid diimide. 2,7-Bis(4,4,5,5-tetramethyl-1,3,2-dioxaborolan-2-yl)-9,9’-spirobifluorene (0.114 g, 0.20 mmol), 4,7-dibromobenzo[*c*] thiadiazole (0.030 g, 0.10 mmol), *N,N*’-bis(2-ethylhexyl)-2,6-dibromonaphtalene-1,4,5,8-tetracarboxylic acid diimide (0.065 g, 0.10 mmol) and aqueous K_2_CO_3_ (2 M, 4.0 mL) were mixed in 1,4-dioxane (20 mL). After degassing for 10 min, tetrakis(-triphenylphosphine) palladium (0.012 g, 5 mol%) was added under an argon atmosphere. The reaction mixture was heated at 110 °C for 48 h before cooling to room temperature. The reaction mixture was extracted with dilute aqueous hydrochloric acid and dichloromethane. The organic layers were combined, washed with brine, dried over anhydrous magnesium sulfate. After filtration, the residue was concentrated and poured into methanol and then the precipitate was collected by filtration. The solid was washed by a Soxhlet extraction with methanol and acetone for 24 h, respectively before being dissolved in hot chloroform. The product was dried under a vacuum to give a yield of 65%. The detailed synthetic process of the polymer is shown in Appendix A. More details of the polymerization reaction condition and the structure characterization of the polymer are reported in reference [21].

### 2.2. Synthesis of Ceria Nanoparticles

Cerium (III) nitrate hexahydrate (Ce(NO_3_)_3_·6H_2_O; 434.23 g/mol, 99.5% purity) was purchased from Alfa Aesar. Ammonium hydroxide (NH_4_OH; 35.05 g/mol, 20%–30%) was purchased from Avantor (Radnor, PA, United States). Ethylene glycol (99.5% purity) was purchased from Sigma-Aldrich. These chemicals were used without further purification. Distilled water was used in the synthesis process for nanoparticles.

CeO_2_ nanoparticles were synthesized by the co-precipitation method. The Ce(NO_3_)_3_·6H_2_O is dissolved in 80% ethylene glycol (EG)/water mixture by stirring at 600 rpm at room temperature. When all the nitrate precursors are dissolved, ammonium hydroxide (3 M) is added dropwise while the solution is kept in the same condition. After the addition of the precipitating agent (i.e., NH_4_OH), the mixture was stirred at 30 °C for 21 h. The precipitate was separated by centrifugation at 6000 rpm for 10 min and washed three times with ethanol. The separated solid was dried for 24 h and crushed with mortar and pistil, the CeO_2_ nanoparticles were obtained as powders.

### 2.3. Synthesis of CeO_2_/Polymer Nanoparticle (NP) Composites

Polymer is dissolved in tetrahydrofuran (THF) with a concentration of 0.001 g/mL and sonicated for 30 min. Different amounts of CeO_2_ nanoparticle powders with a 20, 40 and 50 weight percent were added into the polymer solution and the mixture was sonicated for 30 min. The composites were left in the oil bath for 72 h.

### 2.4. Characterization

The NPs were characterized by using the X-ray diffractometer (XRD) with Cu Kα radiation and beamline 01C2 at the National Synchrotron Radiation Research Center (NSRRC), Taiwan. The particle distribution, morphology, and crystal structure are studied by a transmission electron microscope operated at 200 keV (Philips Technai G2 FEI-TEM) (Thermo Fisher Scientific Co. Amsterdam, Netherlands). The ultraviolet–visible (UV–Vis) spectra were recorded using a Jasco V-670 spectrophotometer (Hachioji, Japan). The photoluminescence properties were investigated using a Jasco FP-8500 spectrofluorometer (Hachioji, Japan). The X-ray absorption near-edge fine structure (XANES) measurements at the Ce *L_3_*-edge were performed at room temperature on a Wiggler beamline 17C at the National Synchrotron Radiation Research Center (NSRRC), Taiwan. The monochromator Si (111) crystals were used in Wiggler beamline 17 C. The energy resolution at the Ce *L_3_*-edge (5723 eV) was about 0.4 eV. The XANES spectra at the C *K*-edge were recorded at beamline 20A using total electron yield (TEY) mode at the National Synchrotron Radiation Research Center (NSRRC), Taiwan. The magnetization was measured at room temperature using a vibrating sample magnetometer (VSM) at the Institute of Physics, Academia Sinica, Taiwan.

## 3. Results and Discussion

### 3.1. Microstructure of the Composites

Figure 1 presents the XRD pattern of pristine polymer, CeO_2_ nanoparticles and CeO_2_/polymer composites with different amounts of CeO_2_ nanoparticles. The XRD pattern of CeO_2_ nanoparticles indicates peaks at 2θ about 28.5°, 33.1°, 47.4°, 56.3°, 59.1°, 69.3°, 76.7° and 79.1° which corresponds to the lattice plane of (111), (200), (220), (311), (222), (400), (331) and (420) of CeO_2_ with space group
$Fm\bar{3}m$
(JCPDS file No. 34-0394). No additional peaks corresponding to other phases were observed which confirms the phase purity of the sample. In the pristine polymer, a relatively broad peak centered at 2θ equal to 19.8° was observed, indicating the amorphous nature of the polymer. Moreover, the XRD pattern of CeO_2_/polymer composites contains all peaks from both components which indicates that the efficient blend of two components.

Figure 2 shows TEM images of CeO_2_ nanoparticles and CeO_2_/polymer composites. As can be seen from Figure 2a, the CeO_2_ nanoparticles highly aggregated together due to the small size and Van der Waal’s forces. The size of the cluster is as large as the micron scale. According to selected area electron diffraction (SAED) analysis (inset of Figure 2a), the particles can be identified as CeO_2_. Figure 2b represents the high-resolution TEM (HRTEM) observation, the particles are highly crystallized with the d-spacing of 0.31 nm which is consistent with (111) plane of CeO_2_ nanoparticles. After the composite formation, the aggregation was suppressed. Figure 2c,d is the TEM and HRTEM images of composites in which the weight ratio between CeO_2_ and polymer is 0.4, showing that the CeO_2_ cluster size was reduced significantly. As the amount of CeO_2_ is further increased (weight ratio between CeO_2_ and polymer is 0.5), the degree of dispersion in the polymer is suppressed. The size of the CeO_2_ cluster was then increased again (Appendix A).

The decrease in the CeO_2_ cluster size could be attributed to the interaction between CeO_2_ nanoparticles and polymer. As mentioned earlier, the polymer used here tends to attract electrons. It has been predicted that naphthalene bisimide units of the polymer are electron-deficient moieties that accept an electron from spirobiflourene moieties [21]. Due to the electron-withdrawing character of the imide groups in the polymer, an electron will be withdrawn from CeO_2_ to the naphthalene bisimide moieties of the polymer. Consequently, the hybridization between oxygen atoms at the top surface and the neighboring cerium atoms will be suppressed. Importantly, the difference in the hybridization between oxygen and cerium can be identified easily by the change in the Ce valence. X-ray absorption spectroscopy (XAS) is a unique method that provides electronic structural information on the orbital symmetry and spin state of the materials and was thus utilized to determine the C hybridization as well as the Ce charge statecaused by the composite formation.

The C *K*-edge XANES of the composites are shown in Figure 3. The main peak at 287.2 eV (peak A) is attributed to the transitions from C 1s to the unoccupied states of C=O π* characters. The peak intensity reflects the unoccupied state of the π* character. After the attachment of CeO_2_, the polymer peak intensity decreased significantly, indicating that more electrons transfer into the polymer. As for the peak located at 290.4 eV (peak B), which correspond to carbon atoms in polymer attached to hydrogen, nitrogen or other species, shows a clear increase. This suggests that the interfacial C–O–Ce bonding is formed for the CeO_2_/polymer composites.

The variation in the Ce charge state in ceria between the composites and pure CeO_2_ was investigated using XANES of the Ce-*L* edge. The normalized XANES spectra of Ce *L*_3_ edge of composites containing different ceria nanoparticle concentration are shown in Figure 4. To determine the Ce valance, an arctangent function was subtracted from each to exclude the edge jump. The XANES spectra of each sample were then fitted with five Gaussian functions. The background subtraction method and assignment of peaks was reported in previous literature [22]. Among these, the shoulder-like feature at about 5727–5729 eV (component C) is contributed by the trivalent Ce (in a final 4f^1^(5d6s)^4^ state). Therefore, the Ce^3+^ concentration in the CeO_2_ matrix (Ce^3+^/(Ce^3+^ + Ce^4+^)) can be then expressed as the ratio
$C^{{Ce}^{3+}}=\frac{I_{C}}{I_{T}}$. Where I_C_ is the deconvoluted peak C intensity and I_T_ is the intensity sum of peaks A, B and C [23]. According to this estimation, the Ce^3+^ concentration (the ratio of the concentration of Ce^3+^/(Ce^3+^ + Ce^4+^)) was about 10% in the raw CeO_2_ nanoparticles. For the composite, whereas the weight ratio between CeO_2_ and polymer is 0.2 and 0.4, the Ce^3+^ concentration decreases to 7.5% and 7%, respectively. The decrease in the Ce^3+^ concentration confirms that electrons transfer from CeO_2_ into the polymer.

The above XANES analysis of the C *K* edge and Ce *L* edge demonstrated the charge transfer between CeO_2_ nanoparticles and the polymer, which results in the increase in electrons in the C=O of polymer and the decrease of hybridization between Ce and O in CeO_2_ nanoparticles. This implies that more Ce^3+^ was induced at the particle surface. It is worth noting that among composites with different amounts of CeO_2_ nanoparticles, the decrease in the Ce^3+^ concentration is almost the same. This suggests that the interaction occurs only at the CeO_2_ nanoparticle surface.

### 3.2. Optical Properties of the Composites

The original characteristics of both components would be affected by the change in the electronic structure as mentioned above. The UV–Vis spectra of the ceria, polymer and composites are shown in Figure 5. The maximum absorption peak is observed at 350 nm for the ceria nanoparticles which arise from the electron transition from the 2*p* valence band (VB) of O^2−^ to the 4*f* conduction band (CB) of Ce^4+^ in CeO_2_. For the polymer, a broad absorption band was observed possibly due to the π-π* transition and the intramolecular charge transfer between spirobifluorenes and naphthalene bisimides [21]. It should be noted that the absorption band of the composites is broader than that of the polymer. When polymer is incorporated with CeO_2_ nanoparticles the polymer absorption is enhanced and the absorption edge shifts to the longer wavelength. This broad absorption and longer absorption edge can be attributed to the charge transfer from the ceria nanoparticles into naphthalene bisimide moieties of the polymer as demonstrated by XAS analysis in the previous section.

The optical band gap can be determined using Tauc’s model using the UV–Vis absorption spectra. Tauc’s model is given by:(1)$$\alpha h\upsilon =a{\left(h\upsilon -E_{g}\right)}^{n}.$$
where α is absorption coefficient, hυ is photon energy, *a* is a constant and n is the index characterizing the nature of electronic transition causing optical absorption. n can take on values of 3, 2, 3/2, or 1/2, corresponding to indirect prohibited, indirect permitted, directly prohibited, and directly permitted transitions, respectively. A graph plotted between (αhυ)^1/n^ as ordinate and hυ as abscissa. The extrapolation of the linear part of the graph to (αhυ)^1/n^ = 0 (Tauc’s plot) gives the optical band gap. Accordingly, the optical band gap is 2.97, 2.02 and 1.70 eV for CeO_2_ nanoparticles, polymer, and composite in which the ratio between CeO_2_ to polymer is 0.4, respectively. The change in the composite band gap indicates the intramolecular charge transfer between spirobifluorenes and naphatalene naphthalene bisimides that leads to a change in the band structure.

Photoluminescence (PL) is a remarkable method to probe certain structural aspects and provide information at short and medium-range, where the degree of local order such as structurally inequitable sites can be distinguished by their different types of electronic transitions and are linked to specific structural arrangements. The structural and electronic order or disorder effects and the nature of bonding in a conjugated polymer have a key impact on the optical property. Figure 6 shows the room-temperature PL spectra of the studied materials excited at 441 nm. No emission peak for CeO_2_ nanoparticles can be observed. A broad emission band from 550 nm to 750 nm with a maximum emission peak (631 nm) was observed for the polymer. The excitons were created by photoexcitation in which the holes and electrons are located in the highest occupied molecular orbital (HOMO) and lowest unoccupied molecular orbital (LUMO) of the polymer, respectively. The electron and hole recombination is responsible for this emission peak. After the composite formation, the fluorescence quenching, as well as red shifting of the emission maximum, is observed. The fluorescence quenching may indicate the intramolecular charge transfer that takes place between the two species as has been confirmed by XANES. The emission peak of the composites shifted from 631 nm to 643 nm. This redshift suggests that there is a change in the polymer electronic structure as well as the chemical environment, which was demonstrated by UV–Vis spectra and XANES of the C-*K* edge of the polymer. This shift may arise due to the reduction in bandgap and the variation in the interface between the CeO_2_ nanoparticles and the polymer.

### 3.3. Magnetic Property of the Composites

M-H curves of the composites (polymer with 40 wt% ceria NPs), pristine polymer and pure ceria NPs measured at room temperature were shown together in Figure 7. All samples are ferromagnetic. The saturation magnetization value (M_s_) of the CeO_2_ is 0.008 emu/g (inset in Figure 7), which is close to those in the literature. In the ceria, the ferromagnetism origin is attributed to the presence of defects, including the cerium vacancy and oxygen vacancy in the structure [24,25]. Several models have been proposed to explain the exchange mechanism [26,27,28]. As for the polymer, the value of M_s_ was as high as 0.3 emu/g. The ferromagnetism of the polymer may be due to the formation of delocalized unpaired electrons on the conjugated polymer owing to the topology. These unpaired electrons make spin-spin interactions during charge transfer between the spirobifluorene and naphthalene bisimide. This interaction may be the cause of the ferromagnetism in the polymer. The large value of H_c_ in the polymer can be related to the topology.

In the polymer composite with 40 wt% CeO_2_ NPs, the value of M_s_ was 0.2 emu/g. It is lower than that of pure polymer. According to the magnetic behavior of pristine polymer and ceria NPs, the decrease in magnetization could be attributed to the lesser content in the polymer. However, it is worth mentioning that the M_s_ of composite is higher than the linear combination of both components, demonstrating the magnetism of both components changed by interface interactions. For ceria NPs, it has been demonstrated that within a wide range of defect concentrations, the value of M_s_ is proportional to the Ce^3+^ concentration at the surface. As shown in the previous section, the Ce^3+^ concentration is less after the composite formation. Based on the above results, the decrease in CeO_2_ ferromagnetic contribution is predicted. In other words, the polymer magnetic response became stronger in the composite. Both polymer conjugation and delocalization was affected by the electron transfer. Similar results were reported by B. Yang et al. They observed that the saturation ferromagnetism of polymer P3HT mixed with phenyl-C61-butryic acid methyl ester (PCBM) was about 0.65 emu/g and suggested that the ferromagnetism origin is associated with P3HT crystallization and the charge transfer between P3HT and PCBM [29]. Accordingly, this study indicated that the magnetism of polymer composite could be further tuned by adjusting the content of each component, and also the interaction at the interface.

## 4. Conclusions

Polymer composite with CeO_2_ NPs was successfully produced using an ex situ preparation method. This method has an advantage over the in-situ method as it is free from the unnecessary solid remaining in the hybrid and it is simple and used for large-scale production. TEM and XAS analysis results showed that the nanoparticles are well dispersed in the polymer matrix with charge transfer that occurred between the two components. The difference in band structure resulted in a broader absorption peak than the individual components UV–Vis spectra. Photoluminescence revealed the fluorescence quenching and red shifting of the composite peak compared to that of the pristine polymer. The difference in electron hybridization and localization also affected the magnetic response of both components. The magnetization of the pristine polymer was found to be enhanced.

## Figures and Tables

**Figure 1 nanomaterials-09-01534-f001:**
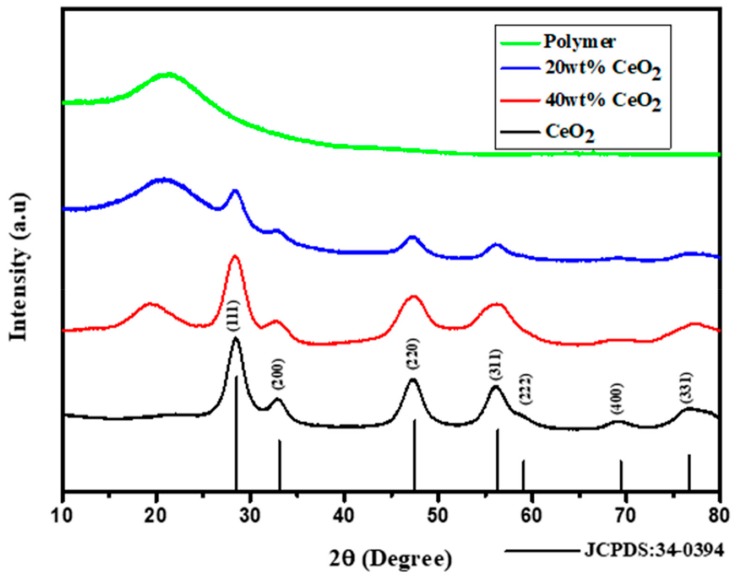
X-ray diffraction (XRD) pattern of polymer composite with 20 wt% and 40 wt% CeO_2_ nanoparticles (NPs).

**Figure 2 nanomaterials-09-01534-f002:**
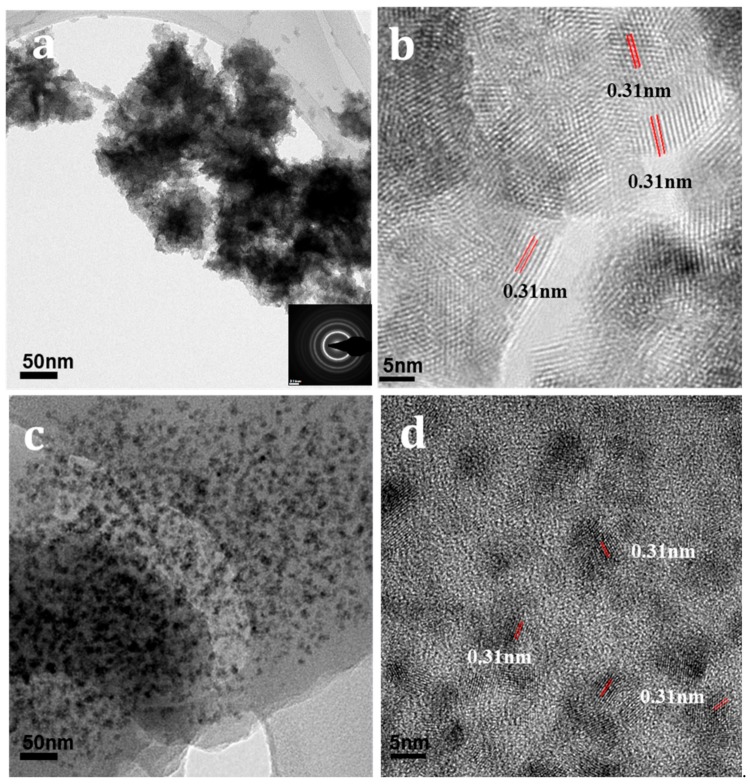
(**a**) Transmission electron microscope (TEM) image and (**b**) high-resolution TEM (HRTEM) of CeO_2_ nanoparticles. (**c**) TEM image and (**d**) HRTEM image of polymer composite with 40 wt% CeO_2_ nanocomposites.

**Figure 3 nanomaterials-09-01534-f003:**
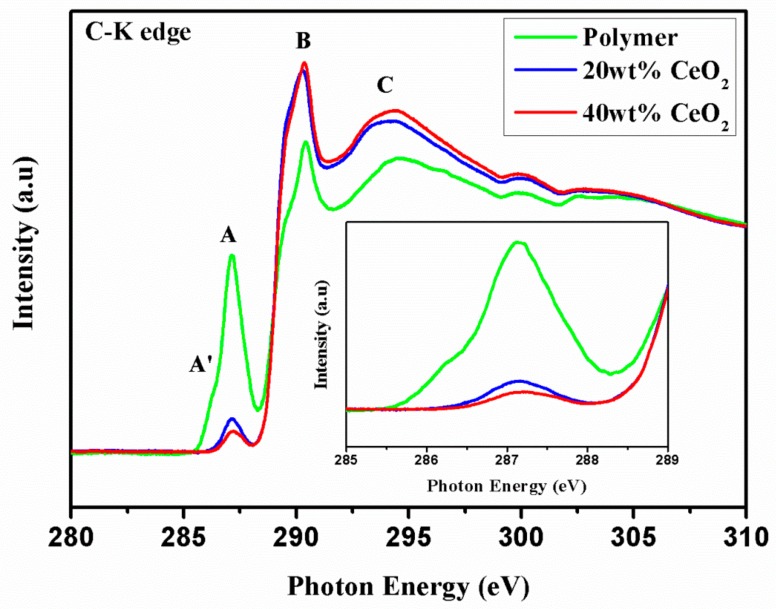
C-K edge X-ray absorption near-edge fine structure (XANES) of polymer, polymer composite with 20 wt% and 40 wt% CeO2 nanoparticles (inset figure shows the magnified peak A).

**Figure 4 nanomaterials-09-01534-f004:**
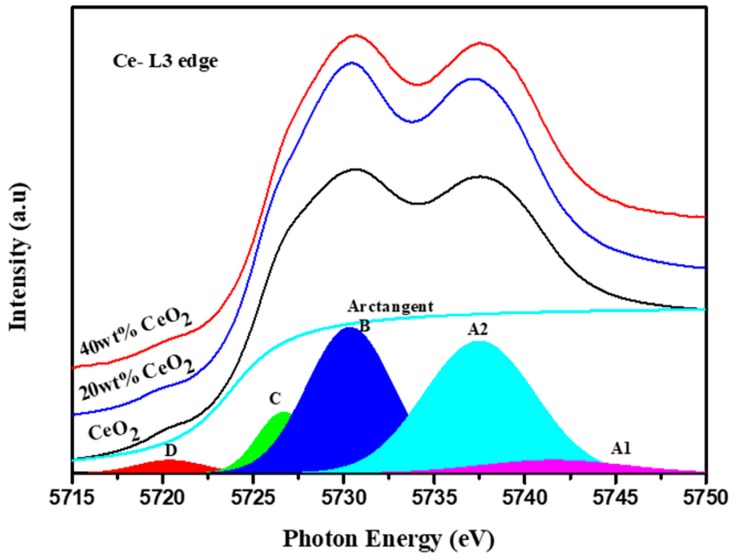
XANES of Ce L edge of CeO_2_, polymer composite with 20 wt% and 40 wt% CeO_2_ nanoparticles.

**Figure 5 nanomaterials-09-01534-f005:**
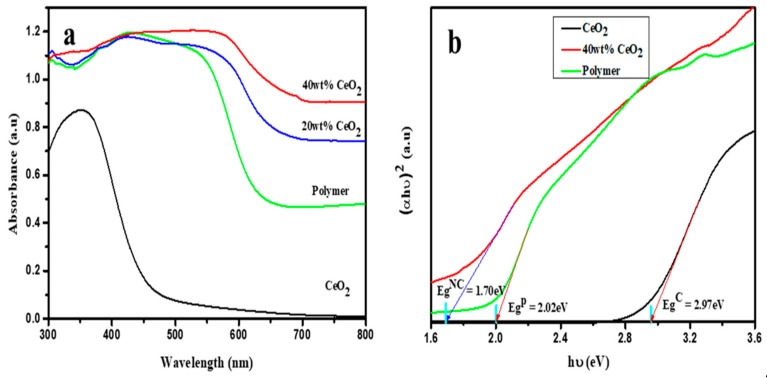
(**a**) Ultraviolet–visible (UV–Vis) spectra of CeO_2_, polymer, polymer composite with 20 wt% and 40 wt% CeO_2_ nanoparticles. The determination of band gap is shown in (**b**).

**Figure 6 nanomaterials-09-01534-f006:**
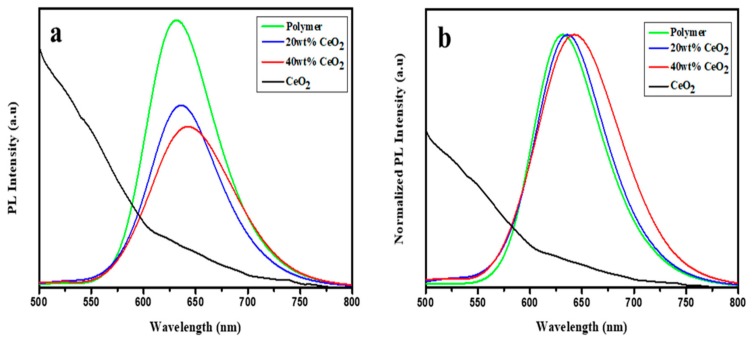
(**a**) The photoluminescence emission spectra and (**b**) the normalized spectra of CeO_2_, polymer, polymer composite with 20 wt% and 40 wt% CeO_2_ nanoparticles.

**Figure 7 nanomaterials-09-01534-f007:**
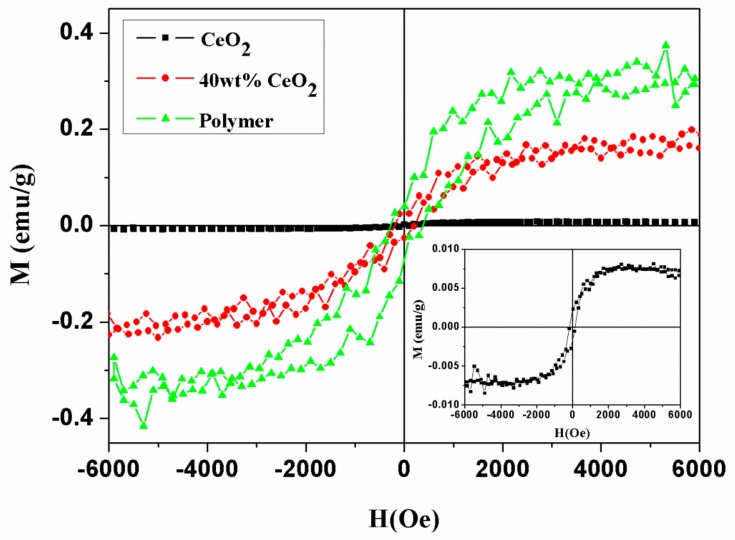
M-H curves of CeO_2_, polymer and polymer composite with 40 wt% CeO_2_ nanoparticles. The magnified M-H curve of CeO_2_ is shown in the inset.

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
