# Peer review of "Interface Interactions in Conjugated Polymer Composite with Metal Oxide Nanoparticles"

_nanomaterials, 2019, doi:10.3390/nano9111534_

Round 1
Reviewer 1 Report
This paper reports on the synthesis and characterization of polymer/CeO2 nanocomposite materials. Hybrid integration of organic and inorganic materials is becoming an important trend in nanotechnology. The proposed material was found to possess a set of interesting properties for various applications. The paper is overall well written with clear presentations. Therefore, the manuscript is acceptable for publication in its present form.
Author Response
Point 1: Moderate English changes required.

Response 1: In the revised manuscript, the English has been re-edited by a native speaker.

Reviewer 2 Report
Polymeric matrix composite enriched with cerium oxide nanoparticles were extensively studied by means of full spectroscopy and microscopy analysis. The results presented show that the hybrid material obtained has different characteristics w.r.t. the single constituents. Information provided are worth of mention, and may be of interest for the skilled reader – the manuscript may be considered for publication after few improvements.
Refer to “J. Phys.: Condens. Matter 19 (2007) 395015” to have fruitful hints about optical properties changes due to nanoparticles-driven interface modulations.
Rows 81-92 are very hard to read: use chemical formulas or, better, provide a schematic picture of the polymer synthesis procedure.
Improve language overall the text by mother tongue revision - ex.: lines 34,35 “The in-situ method is to prepare the nanoparticles directly in the polymer solution, or, otherwise, to carry on the polymerization of monomer containing the nanoparticles.”, lines 39,40 “… even if the dispersion is less efficient compared to the in-situ method and hardly ever keeps a suitable level of stability.”, lines 175,176 “The peak located at 290.4 eV (peak B), which is correspondent to carbon atoms in polymer attached to hydrogen, nitrogen or other species, also shows a clear increase.”, etc. etc.
Author Response
Point 1: Refer to “J. Phys.: Condens. Matter 19 (2007) 395015” to have fruitful hints about optical properties changes due to nanoparticles-driven interface modulations. 

Response 1: Thanks for the valuable comments. This has been added in the revised manuscript as “For instance, Bellucci et al. demonstrated that the optical properties changes were correlated to nanoparticles-driven interface modulations [1].” in lines 29-30.
Point 2: Rows 81-92 are very hard to read: use chemical formulas or, better, provide a schematic picture of the polymer synthesis procedure.
Response 2: We agree with the reviewer that the synthesis process is difficult to read. A schematic picture of the polymer synthesis procedure and chemical formulas for the polymers are added in Fig. S1.
Point 3: Improve language overall the text by mother tongue revision.
Response 3: In the revised manuscript, the English have been re-edited by a native speaker.
